# Dietary Management Tools Improve the Dietary Skills of Patients with T2DM in Communities

**DOI:** 10.3390/nu14214453

**Published:** 2022-10-23

**Authors:** Xueying Chen, Hewei Min, Xinying Sun

**Affiliations:** 1Department of Nursing, Beijing Health Vocational College, Beijing 101101, China; 2School of Public Health, Peking University Health Science Center, Beijing 100191, China

**Keywords:** type 2 diabetes, food management tools, dietary skills, diabetes management

## Abstract

Dietary management is of paramount importance in the prevention and control of type 2 diabetes mellitus (T2DM). This one-year cluster-randomized controlled trial aims to evaluate the effect of dietary management tools on the dietary skills of patients with T2DM. Twenty-two communities were randomly assigned to an intervention group and a control group, and participants in the intervention group received a food guiding booklet (_G_) and a dinner set (_D_). The frequency of dietary management tools usage was collected at baseline and every three months, and different use patterns were identified by a group-based trajectory model. A self-compiled diabetic dietary skills scale and blood glucose were collected at baseline, 3, and 12 months, and a using generalized linear mixed model (GLMM) to explore the influence factors of dietary skills and blood glucose. The finding revealed four dietary tool-usage patterns among the participants: Insist using _G/D_, Give up gradually _G/D_, Give up after use _G_, and Never use _G/D_. GLMM indicated that dietary skills were higher over time (*p* < 0.05), and in participants using the guiding booklet (*p* < 0.001) or dinner set (*p* < 0.001), or with higher education (*p* < 0.001). Additionally, blood glucose were lower among participants with higher dietary skills (*p* = 0.003), higher educational level (*p* = 0.046), and a 3000–5000 monthly income (*p* = 0.026). These findings support using food management tools like the guiding booklet and dinner set as a useful strategy in primary health care centers for individuals with T2DM to increase their dietary skills and blood glucose control.

## 1. Introduction

As one of the fastest-growing diseases worldwide, diabetes is considered a major public health issue. It is estimated that 537 million adults aged 20–79 years are currently living with diabetes, the number of which is predicted to reach 643 million by 2030, and 783 million by 2045. There are 140.9 million adults with diabetes in China, which is a much larger number compared with other parts of the world, indicating the imperative to reduce the health and economic burden of diabetes [1]. Accounting for the vast majority (over 90%) of diabetes globally, type 2 diabetes mellitus (T2DM) can be prevented or delayed by medication and lifestyle behavior changes such as diet, physical activity, and weight management. Dietary modification is one of the cornerstones in the prevention and management of T2DM, which has a body of evidence showing benefits on weight and metabolic control, thus, achieving reduced risk and improvements of outcomes of T2DM [2,3,4]. Dietary components, including carbohydrates, proteins, fats, sugars, macronutrients, micronutrients, vitamins, and minerals, have significant effects on blood glucose levels [5]. Therefore, many healthy dietary patterns based on reasonable ingredient combinations are proven available and effective [2,4,6], such as the Mediterranean diet [7,8], Dietary Approaches to Stop Hypertension (DASH) diet [9,10,11], low-calorie energy deficit diet [12,13,14], low carbohydrate diet [15,16,17], vegan and vegetarian diets [18,19], intermittent fasting and macrobiotic diets [20], and low glycemic index or load dietary patterns [21,22,23,24].

However, knowing what to eat and being on an optimal eating pattern id still a challenge for many people with T2DM. Patients with high dietary skills, such as knowing how to determine the effect of different foods on the glycemic index (GI) and glycemic load (GL) and flexibly making individualized meal plans based on daily caloric needs, may perform better on diet management. Therefore, sufficient dietary education and intervention are supposed to be conducted in primary health care units to improve the dietary skills of people with T2DM. Dietary education requires education, counseling, and diet management, which has shown great improvement in glycemic control and diabetes self-care skills of patients [25,26,27]. With the aim of helping improve acceptance and understanding of dietary education in the diabetic population, many studies have been conducted in various forms, such as face-to-face counseling [28,29,30,31], group education [32,33], peer-supported [34,35], web-based [36,37,38], mobile phone-based [39,40,41], or multifactorial intervention [14,42]. However, we have barely found interventions that included dietary management tools such as graduated tableware and concise food cards, which may provide dietary guidance more directly.

Therefore, the objective of this one-year community-based cluster-randomized controlled trial was to assess the effectiveness of using dietary management tools on the dietary skills and blood glucose control of patients with type 2 diabetes in the community. 

## 2. Materials and Methods

### 2.1. Research Participants

From July to November, 2018, patients with type 2 diabetes were recruited in 22 communities in Shunyi and Tongzhou districts, Bejing, China. Individuals between 18–75 years old were eligible if they had been diagnosed with T2DM, were in permanent residence in Shunyi or Tongzhou district, had not taken any psychotropic drugs prior to enrollment, had not participated in other studies, and agreed to participate. Key exclusion criteria included: a history of type 1 diabetes, gestational diabetes, or secondary diabetes; with severe diabetes complications such as nephropathy, retinopathy, and neuropathy; severe intellectual disability, Alzheimer’s disease, or other mental disorders. A total of 812 patients, which met the required sample size, were enrolled (Figure 1), and written informed consent was obtained from all of them.

### 2.2. Research Design

This one-year community-based cluster-randomized control trial was developed to assess the dietary skills of type 2 diabetic patients and the effect of using dietary management tools on them. The protocol was approved by the Biomedical Ethics Committee of Peking University (IRB00001052-17044). Twenty-two community health centers in Shunyi and Tongzhou districts were randomly assigned to an intervention group and a control group in a 2:1 ratio by a computer-generated list of random numbers. Participants in the intervention group received a food guiding booklet and a dinner set, while patients in the control group received usual care. Data were collected at baseline and 3, 6, 9, 12 months follow-ups. 

### 2.3. Sample Size Calculation

Based on our previous study, with an alpha risk of 0.05, a beta risk of 0.10, an intraclass correlation (ICC) of 0.002, a mean level of HbA1c in patients with T2DM of 7.13%, and a standard deviation (SD) of 1.60%, 200 subjects in each group (157 total before considering the dropout) were expected to be recruited according to the following formula for comparison of two means in cluster randomized controlled trials to detect a decrease in HbA1c to 6.50% after the intervention. If there are 80 subjects in a community, at least 3 communities in each group are needed in each group. As a result, a total of 400 patients in 6 communities are required to be recruited for this program.
(1)$$n=\frac{2{\left(Z_{1-\alpha /2}+Z_{1-\beta }\right)}^{2}\sigma^{2}\left[1+\left(n_{j}-1\right)\rho_{I}\right]}{{\left(\mu_{1}-\mu_{0}\right)}^{2}}$$

### 2.4. Intervention

The control group had basic public health services for T2DM patients (including counselling, follow-up, assessment, health education, referral advice, and medical examination) provided by community doctors [43].

As well as receiving basic public health services, subjects in the intervention group were distributed dietary management tools and educated on how to use them. The dietary management tools included a food guiding booklet and a dinner set (Figure 2). The dinner set was customized for the project, with each set including a graduated rice bowl and a dinner plate. Color-printed food guiding booklet was produced to provide dietary guidance based on the theories of food exchange portion, glycemic index (GI), and glycemic load (GL), the contents of which included the cooked weight and calories per unit mass (50 g/100 g) of various types of food. This booklet used the traffic light method, which has been used in several dietary education programs [44,45,46,47], to distinguish between three GI categories: (1) Green = Low GI and GL = advice to eat, (2) Yellow = Medium GI and GL = advice to eat less, and (3) Red = High GI and GL = advice not to eat. In particular, the dinner set and food in the guiding booklet were the same sizes as their actual liking. Therefore, participants could prepare meals directly according to the amount of food in the book. Patients in intervention groups were educated on dietary management knowledge such as how to calculate total daily energy requirements and how to prepare a meal according to the food guiding book.

### 2.5. Data Collection

Face-to-face follow-ups were conducted by well-trained investigators at baseline and 3, 6, 9, 12 months follow-ups for a detailed evaluation of sociodemographic characteristics, health conditions (including height, weight, BMI, blood glucose level, blood lipids level), and frequency of dietary management tools usage. The dietary skills of participants were collected by the questionnaire at baseline and 3, 12 months follow-ups.

Dietary management tool use was assessed by a questionnaire consisting of seven questions to reflect use frequency, functional utilization, functional evaluation, and reasons for use/non-use of the food guiding booklet and dinner set of patients.

Dietary skill was assessed by a self-compiled diabetic dietary skills scale consisting of five questions including “Determine if you are at a standard weight”, “Calculate your daily calorie needs”, “Develop your personalized recipes”, “Determine the effect of different foods on your blood glucose level”, “Adjust your recipes according to the amount of exercise”. Each question was coded as 1–5 from “unable” to “proficient”. Higher scores were associated with greater dietary management abilities. The Cronbach’s alphas of the scale were 0.701.

### 2.6. Statistical Analysis

The data were uniformly coded and double-parallel entered using Epidata 3.1 (Version 15.0.5, Odense, Denmark). Data analyses were conducted using IBM SPSS Statistics version 24.0 (SPSS Inc., an IBM Company, Chicago, IL, USA). Baseline missing data were filled with multiple imputations. Continuous variables were reported as means ± SD and were compared using Student’s *t* test or one-way ANOVA. Categorical data presented as n(percentage of sample) were compared using χ^2^ tests. To assess the characteristic of dietary management tools used by participants, a group-based trajectory model (GBTM) conducted in Stata (version 14.0, Stata Corporation, College Station, TX) were used to identify various tool use patterns. The number of groups and the trajectory shape (constant, linear, quadratic) were determined by a base model without covariates. The best-fitting model was considered with both a high Bayesian information criterion (BIC) and good explainability. For repeat measurement data, a repeated measures ANOVA was used to explore the changes in blood glucose over time, and a generalized linear mixed model (GLMM) was fitted using a logit link to identify the influence factors (such as sociodemographic characteristics and dietary management tools use patterns) for the dietary skills of patients.

## 3. Results

### 3.1. Feasibility Outcomes

From July to November, 2018, a total of 812 participants were enrolled in the study, excluding those who did not meet the eligibility criteria. There were 708 T2DM patients (87.2%) taking anti-diabetic medications at baseline, and 260 subjects (32.0%) having HbA1c < 6.5% at baseline. The number of participants was 544 in the intervention group and 268 in the control group at baseline. The sociodemographic characteristics and dietary skill scores of the two groups were compared in Table 1, which showed no group differences for gender, age, education, marriage status, monthly income, as well as dietary skill scores.

### 3.2. Trajectory Analysis of Dietary Management Tools Usage Patterns

The BIC results, percentage of the smallest group, and fit parameter estimates of GBTM were presented in the Appendix A. Considering the BIC results and explainability, the usage of food guiding booklet was identified as four patterns among participants in the intervention group (Figure 3A): Insist using (103, 18.9%), Give up gradually (179, 32.9%), Give up after use (55, 10.1%), and Never use (207, 38.1%). Similarly, the usage of the dinner set was presented as three patterns (Figure 3B): Insist using (141, 26.0%), Give up gradually (141, 26.0%), and Never use (309, 56.8%). Insist using referred to the individuals insisting on using dietary management tools from baseline to 12 months. Give up gradually included those who weekly used the food management tools at 3 months follow-up but gradually gave up later. Give up after use referred to those who initially barely used the food guiding booklet but weekly used them at 6 months follow-up and gradually gave up later. And never use represented participants who did not use food management tools throughout the follow-up.

### 3.3. Comparison of Dietary Skill Scores among Different Patterns of Dietary Management Tools Usage

The dietary skill scores among different use patterns of the guiding booklet, dinner set use and the control group were shown in Table 2. There was no group difference among various guiding booklet use patterns and control group at baseline. However, whether insisting or not, participants initially using the guiding booklet had significantly higher dietary skill scores compared with the control group (*p* < 0.001) at the 3 months and 12 months follow-ups. Participants insisting on using the guiding booklet gained the highest dietary skill scores (3 months: 18.08 ± 4.15; 12 months: 19.74 ± 3.75) among other groups, while those who never used the guiding booklet had the poorest dietary skills (3 months: 15.62 ± 4.48; 12 months: 15.44 ± 4.69). 

There were also s difference in dietary skill scores among three different dinner set use groups and control group at baseline and 3, 12 months follow-ups (*p* < 0.001) (Table 2). The dietary skill scores of participants insisting on using the dinner set were 14.77 ± 4.42 at baseline, which was significantly higher than each of the other three groups (never use 12.87 ± 4.26, *p* < 0.001, give up gradually 13.56 ± 4.41, *p* = 0.041, control group 13.42 ± 4.30, *p* = 0.003). Similar results were observed at 3, 12 months follow-ups, where the dietary skill scores decreased with participants giving up using the dinner set quickly or gradually, and were lowest in never using the dinner set group. 

Additionally, whether using dietary management tools or not, participants tended to gain higher dietary skill scores with time passing by. 

### 3.4. The Effects of Dietary Management Tools Using on Dietary Skills

In the generalized linear mixed model, independent variables were dietary management tools use behavior, follow-up time, age, gender, education, marriage status, and monthly income, with changes in dietary skills among baseline, 3 months, and 12 months later as the dependent variable and follow-up time as a random effect. 

The GLMM for the influence factors associated with dietary skills is shown in Table 3 and Table 4, revealing time (*p* < 0.05), guiding booklet using (*p* < 0.001), dinner set using (*p* < 0.001) and education (*p* < 0.001) as significant predictors of dietary skills. Considering the potential interaction, the guiding booklet and dinner set use were separately put into two models.

Table 3 showed that compared with the control group, using the guiding booklet could increase dietary skills, whether insisting or not (Insist using: β 2.292, 95% CI 1.652–2.932; Give up gradually: β 1.569, 95% CI 1.031–2.106; Give up after use: β 1.776, 95% CI 0.898–2.654). Use frequency was positively correlated with the level of improvement of dietary skill scores. In addition, thedietary skills of participants increased from baseline to 12 months follow-ups, but were lower in those who with poor education (Primary school or below: β -1.893, 95% CI −2.463–−1.324; High school: β −1.206, 95% CI −1.815-0.598).

A similar result was observed in dinner set use (Table 4), where participants insisting using the dinner set (β 2.592, 95% CI 2.020–3.165) or giving up gradually (β1.540, 95% CI 0.899–2.181) tended to have higher dietary skill scores compared with the control group, which also increased over time. Additionally, Education was also associated with dietary skills (Primary school or below: β −1.687, 95% CI −2.254–−1.120; High school β −1.095, 95% CI −1.700–−0.490).

### 3.5. The Effects of Dietary Skills on Blood Glucose

The participants’ blood glucose at baseline, a 3-month follow-up, and a 12-month follow-up (M(P25, P75) was 6.9(6.3, 7.7), 6.9(6.3, 7.5), and 6.9(6.4, 7.7). Repeated measures of ANOVA showed that blood glucose had changed over time in the participants (*p* = 0.002). Specifically, blood glucose was lower than baseline at a 3-month follow-up (*p* = 0.002), but increased to roughly the same level as a baseline at a 12-month follow-up (*p* = 0.919) (The results were not reflected in the table). 

Use GLMM to investigate the influence of dietary skills on blood glucose. In the model (Table 5), using blood glucose as the independent variable, dietary skills, follow-up time, age, gender, education, marriage status, and monthly income as the dependent variables. The random effect was the follow-up time. 

According to the GLMM results, dietary skills (*p* = 0.003), education (*p* = 0.046) and monthly income (*p* = 0.026) were all significant predictors of blood glucose among individuals with T2DM. Blood glucose decreased as dietary skills improved (β −0.019, 95% CI −0.032–−0.007). In addition, participants with primary school or below education levels tended to have higher blood glucose than those with college or above education levels (β 0.170, 95% CI 0.003−0.337), and patients with a 3000–5000 monthly income had lower blood glucose than those with 5000 or above monthly income (β −0.160, 95% CI −0.302–−0.019).

## 4. Discussion

These analyses of 812 adults with type 2 diabetes confirm that using dietary management tools like the guiding booklet or dinner set is a helpful practice to improve dietary skills for patients with type 2 diabetes. More significantly, improvements in blood glucose levels were observed in individuals with high dietary skills. Compared to the control group, participants who insisted on using the guiding booklet, gave up gradually, or gave up after use, or with high educational levels, have all gained higher scores in dietary skills, and the same results came for the dinner set use, suggesting that using dietary management tools has a subtle influence on improving and maintaining a scientific diet pattern for people with type 2 diabetes whether they stick with them or not. Furthermore, higher dietary skills, higher education, and a moderate monthly income were all positive predictors of developed blood glucose control. Since the food guiding booklet and dinner sets are economical, and patients who know the composition of food through them can manage dietary patterns themselves, providing food management tools can be a promising dietary intervention in primary health centers for people with type 2 diabetes. 

Certain dietary patterns have been proven to be associated with the onset of type 2 diabetes [9,48], and dietary management is of paramount importance in the prevention and remission of type 2 diabetes [6,49]. Several studies have shown some specific dietary patterns associated with an increased risk of T2DM in the Chinese population, including Western dietary patterns (high in red meats, poultry and organs, processed and cooked meat, eggs, seafood, cheese, fast foods, snacks, chocolates, alcoholic beverages, and coffee) [50], meat patterns (high in meat, animal pluck, fish, and seafood and low in fruits and dairy products) [51], grain patterns (high in grains) [51], and junk food patterns (high in fried food, soft drinks, and desserts) [52], suggesting that there are some commonalities in food choice and eating habits in patients with type 2 diabetes in China, thus, dietary interventions should be implemented in diabetic management. A body of studies has assessed the effects of specific dietary patterns interventions on the remission of T2DM, such as the Mediterranean diet [7,8], DASH diet [9,10,11], low-calorie energy deficit diet [12,13,14], low carbohydrate diet [15,16,17], vegan and vegetarian diets [18,19], intermittent fasting and macrobiotic diets [20], and low GI or GL patterns [21,22,23,24]. Among those, several evidence-based studies have indicated that GI and GL are substantial food markers predicting the development of T2DM for persons of East Asian ancestry [22,23]. In addition, a cost–benefit analysis suggests GI or GL dietary education would produce significant potential cost savings in national healthcare budgets [23]. A collection of guidelines recommended the application of GI education in medical nutrition therapy for the diabetic population [53,54,55,56]. However, to our knowledge, few dietary inventions using food management tools have been conducted in T2DM GI education so far. Yin. et al. [57] conducted a community-based lifestyle intervention program, Pathway to Health (PATH), which included food management tools like an oil and salt measuring cup and recipes, showing favorable changes in weight loss, self-reported diets, and physical activity in Chinese women at risk for diabetes. Grant et al. [47] developed a GI education workshop and provided two evidence-based education materials: The Low Glycemic Index Food Substitution List and The Low Glycemic Index Recipe Booklet, demonstrating an increase in GI knowledge and dietary GI in a relatively short period of time in people living with T2DM. Likewise, our study further indicated the validity of food management tools like food guiding booklets and dish sets on dietary knowledge increase and skills development in patients with T2DM, which are easy to understand and follow for patients, thus, to some extent, overcoming the four barriers to GI education utility identified by educators [47,58]. 

Health literacy is defined as “the degree to which individuals have the capacity to obtain, process, and understand basic health information and services needed to make appropriate health decisions” [59], which is an important positive factor in improving disease progression. T2DM health literacy includes a series of important skills that facilitate self-management and receiving medical services. This includes cultural and conceptual knowledge; the ability to listen and speak (for example, following verbal directions); the ability to write and read (for example, reading and understanding labels on pill bottles, medical directions, education brochures, and other T2DM-related information), and the ability to understand and use numbers (for example, comprehending and following medication dosing, test results, food intake, and exercise levels) [59]. Previous research has shown the relationship between T2DM health literacy, self-management behaviors, and diabetes-related outcomes, such as medication adherence [60], physical activity [61,62], healthy diet [63], self-monitoring of blood glucose [64], and glycemic control [59,61,62]. As a part of T2DM health literacy, improving dietary knowledge and skills may change eating patterns, which in turn can affect glycemic control [65,66,67]. More importantly, our results revealed that better dietary skills were associated with improved blood glucose. Therefore, providing dietary management tools to people with T2DM may have great potential for blood glucose control. Moreover, our study showed that improving the education of people with T2DM may be beneficial in the development of dietary skills and blood glucose control. In addition, increased dietary skill scores of participants both in the intervention group and control group, as time went by, indicating that basic medical care in primary health centers, as well as self-identification of diabetes, also benefits the improvement of dietary knowledge and skills of patients with T2DM. 

This study has several limitations. First, further research is necessary to investigate whether the decrease in blood glucose associated with better dietary skills is the result of a change in dietary patterns after using dietary management tools. Second, the results of the self-compiled diabetic dietary skills scale would likely make our findings an overestimate of the changes in learning and transfer for dietary management in participants. Moreover, exploring the influence factors of different tool-use patterns in people with type 2 diabetes is also an interesting topic, such as personality traits.

Despite these limitations, this is a rare study to our knowledge to examine the association of the food guiding booklet and dinner set use with dietary skills and blood glucose control development in T2DM patients in China. Our findings suggest that dietary education added with food management tools may be a promising and economical strategy in primary health care centers to develop dietary knowledge and skills and maintain healthy diet patterns, thus, achieving partial diabetes remission in patients with type 2 diabetes.

## Figures and Tables

**Figure 1 nutrients-14-04453-f001:**
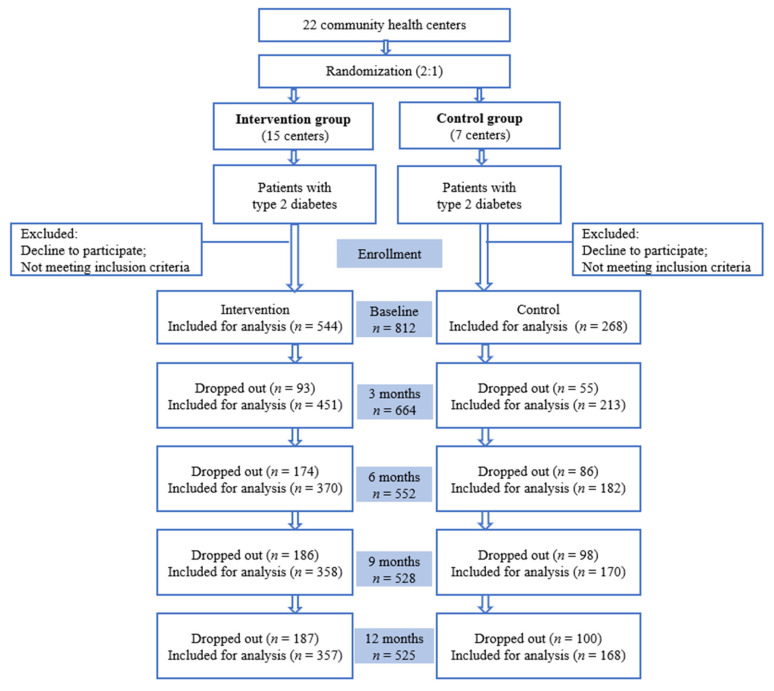
Trial profile.

**Figure 2 nutrients-14-04453-f002:**
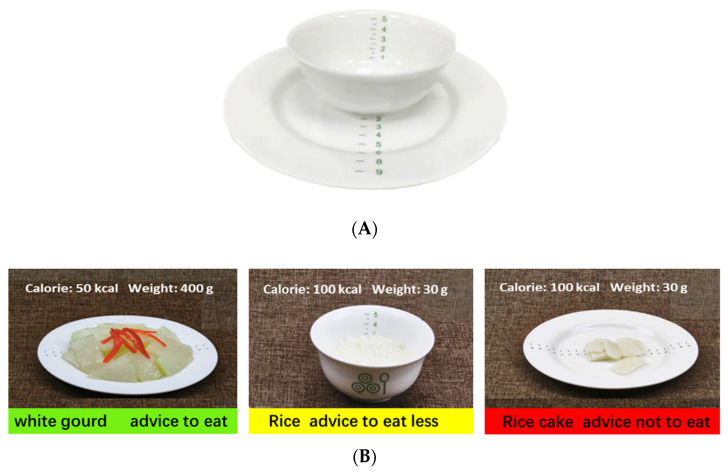
The food management tools. (**A**). The dinner set; (**B**). Contents of the guiding booklet.

**Figure 3 nutrients-14-04453-f003:**
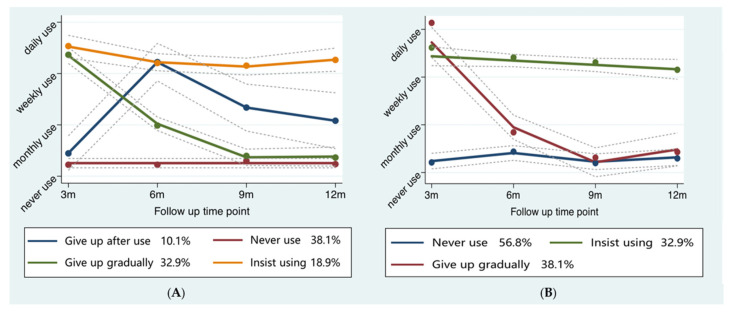
Trajectory analysis of food management tools usage patterns. (**A**) Food guiding booklet usage; (**B**) Dinner set usage.

**Table 1 nutrients-14-04453-t001:** Baseline characteristics of participants in the intervention group and control group.

Characteristics		Intervention (*n* = 544)	Control (*n* = 268)	$t/\chi^{2}$	*p*
Gender ^1^				0.345	0.557
	Male	254 (46.7%)	131 (48.9%)		
	Female	290 (53.3%)	137 (51.1)		
Age ^2^		58.5 ± 8.0	59.4 ± 6.4	1.895	0.059
Education ^1^				4.316	0.116
	Primary school or below	296 (54.4%)	159 (59.3%)		
	High school	141 (25.9%)	72 (26.9%)		
	College or above	107 (19.7%)	37 (13.8%)		
Marriage status ^1^				1.390	0.238
	Married	506 (93.0%)	255 (95.1%)		
	Unmarried	38 (7.0%)	13 (4.9%)		
Monthly income ^1^				3.915	0.271
	<1500	32 (5.9%)	25 (9.3%)		
	1500~	123 (22.6%)	59 (22.0%)		
	3000~	234 (43.0%)	117 (43.7%)		
	5000~	155 (28.5%)	67 (25.0%)		
Dietary skill scores^2^		13.4 ± 4.4	13.4 ± 4.1	−0.035	0.972

^1^ Categorical variable were shown as means ± SD. ^2^ Continuous variable were shown as n (percentage of sample).

**Table 2 nutrients-14-04453-t002:** Comparison of dietary skill scores among different subgroups (means ± SD).

Dietary Management Tools	Group	Baseline	3 Months Follow-Up	12 Months Follow-Up
Guiding book	Give up gradually	13.66 ± 4.43	17.79 ± 4.04	17.73 ± 4.18
	Give up after use	14.17 ± 4.18	16.11 ± 3.86	19.39 ± 4.40
	Never use	12.91 ± 4.31	15.62 ± 4.48	15.44 ± 4.69
	Control group	13.42 ± 4.11	14.85 ± 4.18	15.92 ± 4.86
	F	2.028	15.631	16.444
	*p*	0.089	<0.001	<0.001
Dinner set	Insist using	14.77 ± 4.42	18.40 ± 4.04	19.64 ± 3.76
	Give up gradually	13.56 ± 4.41	17.58 ± 4.56	17.85 ± 4.71
	Never use	12.87 ± 4.26	15.88 ± 4.17	15.98 ± 4.58
	Control group	13.42 ± 4.30	14.85 ± 4.18	15.92 ± 4.48
	F	6.159	20.952	18.216
	*p*	<0.001	<0.001	<0.001

**Table 3 nutrients-14-04453-t003:** The influence of the guiding booklet use on dietary skills.

Parameter	Β (95% CI)	S.E.	*p*
Use behavior			
Insist using	2.292 (1.652~2.932)	0.326	<0.001
Give up gradually	1.569 (1.031~2.106)	0.274	<0.001
Give up after use	1.776 (0.898~2.654)	0.448	<0.001
Never use	−0.219 (−0.705~0.268)	0.248	0.379
Control group	-		
Follow-up time point			
Baseline	−3.370 (−3.848~−2.892)	0.244	<0.001
3 months	−0.652 (−1.148~−0.156)	0.253	0.010
12 months	-		
Gender			
Male	−0.367 (−0.758~0.024)	0.199	0.066
Female	-		
Age	0.008 (−0.017~0.033)	0.013	0.538
Education			
Primary school or below	−1.893 (−2.463~−1.324)	0.290	<0.001
High school	−1.206 (−1.815~−0.598)	0.310	<0.001
College or above	-		
Marriage status			
Married	0.641 (−0.162~1.444)	0.409	0.117
Not married	-		
Monthly income			
<1500	−0.708 (−1.559~0.143)	0.434	0.103
1500~	−0.214 (−0.795~0.368)	0.296	0.471
3000~	−0.314 (−0.804~0.176)	0.250	0.209
5000~	-		
Intercept	16.920 (15.070~18.770)	0.943	<0.001

**Table 4 nutrients-14-04453-t004:** The influence of the dinner set use on dietary skills.

Parameter	Β (95% CI)	S.E.	*p*
Use behavior			
Insist using	2.592 (2.020–3.165)	0.279	<0.001
Give up gradually	1.540 (0.899–2.181)	0.327	<0.001
Never use	0.048 (−0.405–0.501)	0.231	0.836
Control group	-		
Follow-up time point			
Baseline	−3.366 (−3.845–2.888)	0.244	<0.001
3 months	−0.610 (−1.107–0.113)	0.253	0.016
12 months	-		
Gender			
Male	−0.353 (−0.742–0.036)	0.198	0.075
Female	-		
Age	0.007 (−0.018–0.032)	0.013	0.595
Education			
Primary school or below	−1.687 (−2.254–1.120)	0.289	<0.001
High school	−1.095 (−1.700–0.490)	0.308	<0.001
College or above	-		
Marriage status			
Married	0.748 (−0.054–1.550)	0.409	0.067
Not married	-		
Monthly income			
<1500	−0.820 (−1.670–0.029)	0.433	0.058
1500~	−0.234 (−0.815–0.347)	0.296	0.430
3000~	−0.365 (−0.853–0.123)	0.249	0.142
5000~	-		
Intercept	16.754 (14.916–18.592)	0.937	<0.001

**Table 5 nutrients-14-04453-t005:** The influence of dietary skills on blood glucose.

Parameter	β(95% CI)	S.E.	*p*
Dietary skills	−0.019 (−0.032–0.007)	0.006	0.003
Follow-up time point			
Baseline	−0.058 (−0.210–0.093)	0.078	0.451
3 months	−0.050 (−0.197–0.096)	0.075	0.503
12 months	-		
Gender			
Male	−0.048 (−0.160–0.064)	0.057	0.403
Female	-		
Age	0.001 (−0.006–0.008)	0.004	0.740
Education			
Primary school or below	0.170 (0.003–0.337)	0.085	0.046
High school	0.126 (−0.052–0.303)	0.090	0.165
College or above	-		
Marriage status			
Married	0.105 (−0.127–0.337)	0.118	0.375
Not married	-		
Monthly income			
<1500	0.102 (−0.148–0.351)	0.127	0.424
1500~	−0.113 (−0.280–0.054)	0.085	0.185
3000~	−0.160 (−0.302–0.019)	0.072	0.026
5000~	-		
Intercept	7.253 (6.692–7.815)	0.286	<0.001

## Data Availability

Not applicable.

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
