# Peer review of "Dietary Management Tools Improve the Dietary Skills of Patients with T2DM in Communities"

_nutrients, 2022, doi:10.3390/nu14214453_

Round 1

Reviewer 1 Report

The study by Chen is of interest. In fact, it adds new information on dietary management. 

Specifc comments

1. Exact diabetes treatments is each group should be specified.

2 Information on glycemic control at the beginning and at the end of the study should be given. 

3. Exact p-values should be reported in the abstract.

4. How sample -size was estimated should be specified.

5. The authors should explain what they mean with the sentence "severe diabetes complications"

6. Did the authors monitor complications during the follow up? In fact, patient education may affect complications (Gazzaruso C. , Endocrine 2016).

Author Response

Point 1: Exact diabetes treatments is each group should be specified.

Response 1: Thank you very much for your comments. Participants in this study had dietary management tools intervention plus T2DM basic public health services (intervention group) or T2DM basic public health services. The T2DM basic public health services included T2DM counselling, follow-up, assessment, health education, referral advice, and medical examinations provided by community doctors (in “2.4. Intervention”, lines 104-106). During the study, if participants’ conditions became worse or they developed complications, we recommended further clinical treatment. We did not provide any other medical treatment.

Point 2: Information on glycemic control at the beginning and at the end of the study should be given.

Response 2: Thank you very much for your comments. According to your comments, we have added the patients' blood glucose levels at baseline, 3-month follow up, and 12-month follow up, and explored the impact of improved dietary skills on glycemic control. Results were shown in “3.5. The effects of dietary skills on blood glucose”.

Point 3: Exact p-values should be reported in the abstract.

Response 3: Thank you very much for your comments. In the abstract, there was more than one P value of follow-up time to the dietary skills (at baseline: PG/D <0.001, at 3-month follow up: PG =0.010,  PD =0.016). We weren’t sure whether we should report all the P values. Therefore, we put P<0.05 in the abstract, line 19, for simplicity.

Point 4: How sample -size was estimated should be specified.

Response 4: Thank you very much for your comments. According to your comments, we added sample size calculation in lines 91-102.

Point 5: The authors should explain what they mean with the sentence "severe diabetes complications".

Response 5: Thank you very much for your comments. According to your comments, we added “with severe diabetes complications such as nephropathy, retinopathy, and neuropathy” in “2.1 Research participants”, lines 73-74.

Point 6: Did the authors monitor complications during the follow up? In fact, patient education may affect complications (Gazzaruso C. , Endocrine 2016).

Response 6: Thank you very much for your comments. We monitored the participants’ complications, and found that there were no severe T2DM complications among individuals during follow-up. According to your comments, we will study the influence of dietary management tools on T2DM complications in further research.

Reviewer 2 Report

In this manuscript, the authors aimed to determine the effectiveness of dietary management tools on dietary skills of T2DM patients

[1] Were all subjects newly diagnosed T2DM or pre-existing T2DM upon recruitment? Were subjects on any anti-diabetic medications. Some T2DM patients on medications could have fasting glucose < 7.0 mmol/l or HbA1c < 6.5%.

[2] For the findings of this study, the authors reported only two self-reported outcomes, namely dietary management tools use and dietary skill. As stated in the section 2.4, nutrition parameters such as BMI, blood glucose level, and blood lipids level were collected at each time point of follow up. Therefore, these data should also be reported to enhance the clinical relevance of this study. It is also to establish the relationship between dietary skills and these nutrition parameters. Otherwise, the existing data may not be really interest of the reader.

Author Response

Point 1: Were all subjects newly diagnosed T2DM or pre-existing T2DM upon recruitment? Were subjects on any anti-diabetic medications. Some T2DM patients on medications could have fasting glucose < 7.0 mmol/l or HbA1c < 6.5%.

Response 1: Thank you very much for your comments. In this study, all subjects were pre-existing T2DM (had been diagnosed T2DM before we recruited). There were 708 T2DM patients (87.2%) taking anti-diabetic medications at baseline, and 260 subjects (32.0%) having HbA1c < 6.5% at baseline. According to your comments, we found that we incorrectly wrote the diagnostic criteria for T2DM as inclusion criteria. We have corrected the error in “2.1 Research participants” to “had been diagnosed with T2DM”.

Point 2:  For the findings of this study, the authors reported only two self-reported outcomes, namely dietary management tools use and dietary skill. As stated in the section 2.4, nutrition parameters such as BMI, blood glucose level, and blood lipids level were collected at each time point of follow up. Therefore, these data should also be reported to enhance the clinical relevance of this study. It is also to establish the relationship between dietary skills and these nutrition parameters. Otherwise, the existing data may not be really interest of the reader.

Response 2: Thank you very much for your comments. According to your comments, we added the patients' blood glucose levels at baseline, 3-month follow up, and 12-month follow up, and explored the impact of improved dietary skills on glycemic control. Results were shown in “3.5. The effects of dietary skills on blood glucose”. Additionally, we also examined the effect of dietary skills on BMI, which was not significant in the GLMM model. Therefore, due to space restrictions, we didn’t put BMI results in this article.

Round 2

Reviewer 1 Report

the review has been done 
